# Reduced-Order $H_\infty$ Filter Design for Singular Fractional-Order Systems

**Ying Guo** [1,2], **Chong Lin** [1,*] and **Bing Chen** [1]

1   Institute of Complexity Science, Qingdao University, Qingdao 266071, China; guoyinglll@163.com (Y.G.);
    chenbing1958@126.com (B.C.)
2   School of Mathematic and Statistics, Zaozhuang University, Zaozhuang 277100, China
*   Correspondence: linchong_2004@hotmail.com

**Abstract:** This paper investigates the problem of reduced-order $H_\infty$ filter design for singular fractional-order systems with order $0 < \alpha < 1$. It provides necessary and sufficient conditions for designs of both reduced-order $H_\infty$ filters and zeroth-order $H_\infty$ filters. When reduced to special cases, the present results are shown to include those in recent works as special cases. Illustrative examples are presented to demonstrate the effectiveness of the results.

**Keywords:** singular fractional-order systems; reduced-order $H_\infty$ filter; admissibility

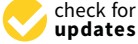



## 1. Introduction

With the development of social industries, there are higher requirements for the establishment of mathematical models. Fractional order models are more accurate in describing the actual systems, which make it easier to study and control these systems. Fractional calculus and fractional-order system (FOS) theories have attracted extensive attention, and the related theories are becoming more and more mature [1–11].

Singular systems include differential equations describing the dynamic characteristics of state variables and algebraic equations describing the static relationship of state variables [12]. The studies of singular fractional-order systems have more practical significance, and the singular FOS theories have the prospect of a very broad application. Many FOS theories have been extended to singular fractional-order systems. Stability is the basis of control systems, including singular FOS. The authors of [13–15] focus on the regularity, impulse-free, and admissibility properties and establish sufficient and necessary conditions for the admissibility for singular FOSs. Reference [16] presents new admissibility conditions of singular FOS with order $1 < \alpha < 2$ expressed in a set of strict linear matrix inequalities (LMI). References [17,18] study the robust stabilization of uncertain singular FOSs. With the development of stability theory of singular FOSs, the study of stabilization has also received high attention. The static and dynamic output feedback stabilization problems of singular FOSs have been considered in [19–21], and the observer-based stabilization for a class of singular FOSs have been considered in [22].

Filtering is one of the most important basic problems in the field of control. The output signal is estimated by a certain filter for the unmeasurable signal inside the system. The classical Kalman filtering theory is based on an accurate mathematical model. It is assumed that the input noise is a strictly Gaussian process or a Gaussian sequence, but the two conditions cannot be met in practice. In the actual modeling, in addition to the stability, the system also needs to meet some performance indicators, for example, $H_\infty$, $L_\infty$ performance index [23–26]. Compared with traditional Kalman filter, the advantage of $H_\infty$ filtering is that one does not need to accurately know the statistical properties of external interference signals. One only needs to assume that the external interference is energy-bounded and that the $H_\infty$ filter has better robustness to the uncertainty in the model. These advantages make $H_\infty$ filtering more widely used, and many scholars apply it to

various complex systems [27–30]. The reduced-order filtering, whose order is lower than that of the system, is a very important issue in many applications [31,32]. For singular FOSs, the reduced-order $H_\infty$ filtering design problem has not been sufficiently solved [33,34]. Therefore, the research on these issues is more challenging, and has more theoretical and practical application value.

In this paper, we investigate the problem of reduced-order $H_\infty$ filter design for singular FOSs with the fractional commensurate order $0 < \alpha < 1$. Our purpose is to design appropriate filters such that the filtering error systems are admissible and the transfer functions from the disturbance to the filtering error output satisfy a prescribed $H_\infty$-norm bound constraint. We provide sufficient and necessary conditions for $H_\infty$ filtering design for singular FOSs. Numerical examples illustrate the effectiveness of the methods.

The paper is organized as follows. In Section 2, we provide the problem formulation and useful lemmas. In Section 3, the main results are presented. Section 4 gives numerical examples to illustrate our proposed results. Section 5 is the conclusion.

**Notation 1.** *Throughout this paper, $X^{-1}$ and $X^T$ denote the inverse and the transpose of $X$, respectively. $X > 0$ $(< 0)$ denotes that $X$ is positive (negative). Given $X \in \mathbb{R}^{n \times m}$, rank $X = r$, the orthogonal complement $X^\perp \in \mathbb{R}^{(n-r) \times n}$ satisfies $X^\perp X = 0$ and $X^\perp X^{\perp T} > 0$. The notation $sym\{R\}$ stands for $R + R^T$. Furthermore, $\mathbb{P}_\alpha^{n \times n} = \{aX + bY, X, Y \in \mathbb{R}^{n \times n}, \begin{bmatrix} X & Y \\ -Y & X \end{bmatrix} > 0, \ a = \sin\frac{\alpha\pi}{2}, b = \cos\frac{\alpha\pi}{2}\}$. The symmetric term in a matrix is denoted by $*$. Matrices, if not explicitly stated, are assumed to have compatible dimensions.*

## 2. Problem Formulation and Preliminaries

In this paper, we study the $H_\infty$ filtering problem for singular FOS. Consider the following singular FOS described by

$$\begin{cases} ED^\alpha x(t) = Ax(t) + B\omega(t) \\ \qquad y(t) = Cx(t) \\ \qquad z(t) = Lx(t) \end{cases} \tag{1}$$

where $x(t) \in \mathbb{R}^n$ is the state vector; $y(t) \in \mathbb{R}^m$ is the measurement; $z(t) \in \mathbb{R}^q$ is the signal to be estimated, and $\omega(t) \in \mathbb{R}^p$ is the disturbance input that belongs to $\mathcal{L}_2[0, \infty)$. The matrix $E \in \mathbb{R}^{n \times n}$ may be singular, $rank(E) = r \leq n$. $A, B, C$, and $L$ are known real constant matrices with appropriate dimensions, $0 < \alpha < 1$. We use the Caputo fractional derivative, which, with order $\alpha$ for a function $f(t)$, is defined as:

$$D^\alpha f(t) = \frac{1}{\Gamma(h - \alpha)} \int_0^t \frac{f^{(h)}(s)}{(t - s)^{\alpha - h + 1}} ds, \tag{2}$$

where $h$ is an integer satisfying $h - 1 < \alpha \leq h$ and $\Gamma(\cdot)$ is the Gamma function.

**Definition 1** ([14])**.** *The triple $(E, A, \alpha)$ or the system $ED^\alpha x(t) = Ax(t)$ is called regular if $det(s^\alpha E - A)$ is not identically zero. It is called impulse-free if $deg(det(sE - A)) = rank(E)$. It is called stable if all the roots of $det(s^\alpha E - A) = 0$ satisfy $|arg(spec(E, A, \alpha))| > \alpha\frac{\pi}{2}$, where $spec(E, A, \alpha)$ is the spectrum of $det(s^\alpha E - A) = 0$. The triple $(E, A, \alpha)$ is said to be admissible if $(E, A, \alpha)$ is regular, impulse-free, and stable.*

The transfer function between $\omega(t)$ and $z(t)$ is

$$G_{\omega z}(s) = L(s^\alpha E - A)^{-1} B. \tag{3}$$

Consider the following filter

$$\begin{cases} D^\alpha \hat{x}(t) = \hat{A}\hat{x}(t) + \hat{B}y(t) \\ \quad \hat{z}(t) = \hat{C}\hat{x}(t) + \hat{D}y(t) \end{cases} \tag{4}$$

where $\hat{x}(t) \in \mathbb{R}^{\hat{n}}$ is the state vector, $\hat{n} < r$, $\hat{z}(t) \in \mathbb{R}^q$ is the estimator of $z(t)$. Matrices $\hat{A}, \hat{B}, \hat{C}$, and $\hat{D}$ are to be determined.

Let $\tilde{x}(t) = [x(t)^T \ \hat{x}(t)^T]^T$, $\tilde{z}(t) = z(t) - \hat{z}(t)$. Then, the filtering error system is described as

$$\begin{cases} \widetilde{E}D^\alpha \tilde{x}(t) = \widetilde{A}\tilde{x}(t) + \widetilde{B}\omega(t) \\ \quad \tilde{z}(t) = \widetilde{C}\tilde{x}(t) \end{cases} \tag{5}$$

where

$$\widetilde{E} = \begin{bmatrix} E & 0 \\ 0 & I \end{bmatrix}, \ \widetilde{A} = \begin{bmatrix} A & 0 \\ \hat{B}C & \hat{A} \end{bmatrix}, \ \widetilde{B} = \begin{bmatrix} B \\ 0 \end{bmatrix}, \ \widetilde{C} = \begin{bmatrix} L - \hat{D}C & -\hat{C} \end{bmatrix}, \tag{6}$$

and the transfer function between $\omega(t)$ and $\tilde{z}(t)$ is

$$G_{\omega\tilde{z}}(s) = \widetilde{C}(s^\alpha \widetilde{E} - \widetilde{A})^{-1}\widetilde{B}. \tag{7}$$

The purpose of this work is to solve the following problem.

**Problem 1.** *Given the singular FOS in (1) and a prescribed $H_\infty - norm$ bound $\gamma > 0$, find a reduced-order $H_\infty$ filter in (4), such that the following two conditions hold.*
*(1) The filtering error system in (5) is admissible.*
*(2) The transfer function $G_{\omega\tilde{z}}(s)$ satisfies $\|G_{\omega\tilde{z}}(s)\|_\infty < \gamma$.*

If the problem has a solution, then we call the reduced-order $H_\infty$ filtering problem is solvable.

Especially when $\hat{n} = 0$, the reduced-order filter in (4) is deformed to

$$\hat{z}(t) = \hat{D}y(t), \tag{8}$$

where $\hat{z}(t) \in \mathbb{R}^q$ is the estimator of $z(t)$ and $\hat{D}$ is to be determined.

For later development, we first introduce some lemmas.

**Lemma 1** ([19]). *The triple $(E, A, \alpha)$ with $0 < \alpha < 1$ is admissible, if and only if there exist $P \in \mathbb{P}_\alpha^{n \times n}$ and $Q$ such that*

$$sym\{APE + AE_0Q\} < 0, \tag{9}$$

*where $E_0 \in \mathbb{R}^{n \times (n-m)}$ is an arbitrarily matrix of full column rank and satisfies $E^T E_0 = 0$.*

**Lemma 2** ([34]). *Given a scalar $\gamma > 0$, the singular FOS in (1) is admissible and the transfer function satisfies $\|G_{\omega z}(s)\| < \gamma$ if and only if there exist matrices $X, Y$ such that:*

$$\begin{bmatrix} E^T X & E^T Y \\ -E^T Y & E^T X \end{bmatrix} = \begin{bmatrix} X^T E & -Y^T E \\ Y^T E & X^T E \end{bmatrix} \geq 0, \tag{10}$$

$$\begin{bmatrix} AY + Y^T A^T & Y^T L^T & B \\ * & -\gamma I & 0 \\ * & * & -\gamma I \end{bmatrix} < 0, \tag{11}$$

*where*

$$\Upsilon = aX - bY, a = \sin\frac{\alpha\pi}{2}, b = \cos\frac{\alpha\pi}{2}. \tag{12}$$

According to the above lemmas, we can easily get the following Lemma 3.

**Lemma 3.** *Given a scalar $\gamma > 0$, the singular FOS in (1) is admissible and the transfer function satisfies $\|G_{\omega z}(s)\| < \gamma$ if and only if there exist matrices $P \in \mathbb{P}_\alpha^{n\times n}$ and $Q$ such that*

$$\begin{bmatrix} sym\{A^T(PE + E_0Q)\} & L^T & (PE + E_0Q)^T B \\ * & -\gamma I & 0 \\ * & * & -\gamma I \end{bmatrix} < 0, \tag{13}$$

*where $E_0 \in \mathbb{R}^{n\times(n-m)}$ is an arbitrary matrix of full column rank and satisfies $E^T E_0 = 0$.*

**Proof.** By Lemma 2, letting $U = \begin{bmatrix} \Upsilon^{-1} & 0 & 0 \\ 0 & I & 0 \\ 0 & 0 & I \end{bmatrix}$ and left- and right-multiplying (11) by $U^T$ and $U$, respectively, (10) and (11) are found to be equivalent to that there exist matrices $X_0, Y_0$, such that

$$\begin{bmatrix} E^T X_0 & E^T Y_0 \\ -E^T Y_0 & E^T X_0 \end{bmatrix} = \begin{bmatrix} X_0^T E & -Y_0^T E \\ Y_0^T E & X_0^T E \end{bmatrix} \geq 0, \tag{14}$$

$$\begin{bmatrix} Y_0^T A + A^T Y_0 & L^T & Y_0 B \\ * & -\gamma I & 0 \\ * & * & -\gamma I \end{bmatrix} < 0, \tag{15}$$

where

$$\Upsilon_0 = \Upsilon^{-1} = aX_0 - bY_0, a = \sin\frac{\alpha\pi}{2}, b = \cos\frac{\alpha\pi}{2}. \tag{16}$$

Using Lemma 1, we obtain the condition (13). □

**Lemma 4** ([29]). *Given a symmetric matrix $\Sigma \in \mathbb{R}^{n\times n}$ and two matrices $\Gamma \in \mathbb{R}^{n\times m}, \Pi \in \mathbb{R}^{k\times n}$, there exists a matrix $\Theta$ to solve the following matrix inequality*

$$\Omega + \Gamma\Theta\Pi + (\Gamma\Theta\Pi)^T < 0, \tag{17}$$

*if and only if the following two conditions are satisfied*

$$\Gamma^\perp \Omega \Gamma^{\perp T} < 0, \quad (\Pi^T)^\perp \Omega (\Pi^T)^{\perp T} < 0. \tag{18}$$

*In this case, the matrix $\Theta$ can be expressed as*

$$\Theta = -S^{-1}\Gamma^T \Delta \Pi^T \Lambda + \Sigma^{1/2} W \Lambda^{1/2} \tag{19}$$

*where matrices $S, L$ and $\Delta$ are free parameters satisfying*

$$S > 0, \quad \| W \| < 1, \quad \Delta = (\Gamma S^{-1}\Gamma^T - \Omega)^{-1} > 0, \tag{20}$$

*and $\Lambda$ and $\Sigma$ are defined by*

$$\Sigma = S^{-1} - S^{-1}\Gamma^T(\Delta - \Delta\Pi^T\Lambda\Pi\Delta)\Gamma S^{-1}, \tag{21}$$

$$\Lambda = (\Pi\Delta\Pi^T)^{-1}. \tag{22}$$

### 3. Main Results

In this part, we will introduce the design methods of reduced-order $H_\infty$ filter and zeroth-order $H_\infty$ filter for singular FOSs.

**Theorem 1.** *The reduced-order $H_\infty$ filtering problem for the singular FOS in (1) is solvable with an $\hat{n}$-th order filter in the form (4) if and only if there exist matrices $P_1 \in \mathbb{P}_\alpha^{n \times n}$, $P_1 + P_2 \in \mathbb{P}_\alpha^{n \times n}$ and Q, $rankP_2 \le \hat{n}$ satisfying*

$$
\begin{bmatrix} sym\{A^T(P_1E + E_0Q)\} & (P_1E + E_0Q)^T B \\ * & -\gamma I \end{bmatrix} < 0, \tag{23}
$$

$$
\begin{bmatrix} C_1 sym\{A(P_1E + P_2E + E_0Q)\}C_1^T & C_1 L^T & C_1(P_1E + P_2E + E_0Q)^T B \\ * & -\gamma I & 0 \\ * & * & -\gamma I \end{bmatrix} < 0. \tag{24}
$$

*where*

$$
C_1 = (C^T)^\perp, E^T E_0 = 0. \tag{25}
$$

*With the feasible solutions $P_1, Q$, and $P_2$, the parameters in (4) are given by*

$$
\begin{bmatrix} \hat{D} & \hat{C} \\ \hat{B} & \hat{A} \end{bmatrix} = -S^{-1}\Gamma^T \Delta \Pi^T \Lambda + \Sigma^{1/2} W \Lambda^{1/2}, \tag{26}
$$

*where $S, W, \Lambda, \Delta$ and $\Sigma$ satisfy (20)–(22) with parameters $\Omega, \Gamma$, and $\Pi$ as follows.*

$$
\Omega = \begin{bmatrix} sym\{A^T(PE + E_0Q)\} & A^T P_{12} & L^T & (PE + E_0Q)^T B \\ * & 0 & 0 & P_{12}^T B \\ * & * & -\gamma I & 0 \\ * & * & * & -\gamma I \end{bmatrix},
$$

$$
\Gamma = \begin{bmatrix} 0 & P_{21}^T \\ 0 & P_{22}^T \\ -I & 0 \\ 0 & 0 \end{bmatrix}, \Pi = \begin{bmatrix} C & 0 & 0 & 0 \\ 0 & I & 0 & 0 \end{bmatrix}. \tag{27}
$$

*In addition, $P_{22} \in \mathbb{P}_\alpha^{\hat{n} \times \hat{n}}$, $P_{12} = aX_{12} + bY_{12}$, $P_{21} = aX_{12}^T - bY_{12}^T$, where $X_{12}, Y_{12} \in \mathbb{R}^{n \times \hat{n}}$, satisfying $P_2 = P_{12} P_{22}^{-1} P_{21}$.*

**Proof.** The matrices in (5) can be decomposed as

$$
\widetilde{A} = \bar{A} + \bar{F} G \bar{H}, \widetilde{C} = \bar{C} + \bar{S} G \bar{H}, \tag{28}
$$

where

$$
\bar{A} = \begin{bmatrix} A & 0 \\ 0 & 0 \end{bmatrix}, \bar{F} = \begin{bmatrix} 0 & 0 \\ 0 & I \end{bmatrix}, G = \begin{bmatrix} \hat{D} & \hat{C} \\ \hat{B} & \hat{A} \end{bmatrix},
$$

$$
\bar{H} = \begin{bmatrix} C & 0 \\ 0 & I \end{bmatrix}, \bar{C} = \begin{bmatrix} L & 0 \end{bmatrix}, \bar{S} = \begin{bmatrix} -I & 0 \end{bmatrix}, \tag{29}
$$

□

By Lemma 3, the filtering error system in (5) is admissible and the transfer function $G_{\omega\tilde{z}}(s)$ of the error system satisfies $\|G_{\omega\tilde{z}}(s)\|_\infty < \gamma$ if and only if there exist matrices $\widetilde{P} \in \mathbb{P}_\alpha^{(n+\hat{n})\times(n+\hat{n})}, \widetilde{Q}$, such that

$$\begin{bmatrix} sym\{\widetilde{A}^T(\widetilde{P}\widetilde{E} + \widetilde{E_0}\widetilde{Q})\} & \widetilde{C}^T & (\widetilde{P}\widetilde{E} + \widetilde{E_0}\widetilde{Q})^T\widetilde{B} \\ * & -\gamma I & 0 \\ * & * & -\gamma I \end{bmatrix} < 0, \tag{30}$$

where $\widetilde{E_0}$ is an arbitrary matrix of full column rank and satisfies $\widetilde{E}^T\widetilde{E_0} = 0$. From (30), we obtain that

$$sym\{\widetilde{A}^T(\widetilde{P}\widetilde{E} + \widetilde{E_0}\widetilde{Q})\} < 0, \tag{31}$$

which means the matrix $\widetilde{P}\widetilde{E} + \widetilde{E_0}\widetilde{Q}$ is nonsingular and $\widetilde{P}, \widetilde{E_0}, \widetilde{Q}$ can be decomposed as

$$\widetilde{P} = \begin{bmatrix} P & P_{12} \\ P_{21} & P_{22} \end{bmatrix}, \widetilde{E_0} = \begin{bmatrix} E_0 \\ 0 \end{bmatrix}, \widetilde{Q} = \begin{bmatrix} Q & 0 \end{bmatrix}, \tag{32}$$

where the decomposition is compatible with $\bar{A}$, and $E^TE_0 = 0$. Due to $\widetilde{P} = \begin{bmatrix} P & P_{12} \\ P_{21} & P_{22} \end{bmatrix} = a\begin{bmatrix} X & X_{12} \\ X_{21} & X_{22} \end{bmatrix} + b\begin{bmatrix} Y & Y_{12} \\ Y_{21} & Y_{22} \end{bmatrix}, \begin{bmatrix} \widetilde{X} & \widetilde{Y} \\ -\widetilde{Y} & \widetilde{X} \end{bmatrix} > 0$, we can obtain $\begin{bmatrix} X & Y \\ -Y & X \end{bmatrix} > 0$, $\begin{bmatrix} X_{22} & Y_{22} \\ -Y_{22} & X_{22} \end{bmatrix} > 0$, $P = aX + bY, P_{12} = aX_{12} + bY_{12}, P_{21}^T = aX_{12} - bY_{12}$, and $P_{22} = aX_{22} + bY_{22}$. By Lemma 1 in [11], $P$ and $P_{22}$ are invertible. Let $U = \begin{bmatrix} I & 0 \\ -P_{22}^{-1}P_{21} & I \end{bmatrix}$. By left- and right-multiplying $\widetilde{P}$ by $U^T$ and $U$, respectively, it is easy to deduce that $P - P_{12}P_{22}^{-1}P_{21} = a\hat{X}_{11} + b\hat{Y}_{11}, \begin{bmatrix} \hat{X}_{11} & \hat{Y}_{11} \\ -\hat{Y}_{11} & \hat{X}_{11} \end{bmatrix} > 0, P - P_{12}P_{22}^{-1}P_{21}$ is invertible [10]. Since

$$\widetilde{P}\widetilde{E} + \widetilde{E_0}\widetilde{Q} = \begin{bmatrix} PE + E_0Q & P_{12} \\ P_{21}E & P_{22} \end{bmatrix}, \tag{33}$$

we have

$$(\widetilde{P}\widetilde{E} + \widetilde{E_0}\widetilde{Q})^{-1} = \begin{bmatrix} R & R_{12} \\ R_{21} & R_{22} \end{bmatrix}, \tag{34}$$

where $R = [(P - P_{12}P_{22}^{-1}P_{21})E + E_0Q]^{-1}$.

On the other hand, with (29), the LMI in (30) can be expressed as

$$\begin{bmatrix} sym\{(\bar{A} + \bar{F}G\bar{H})^T(\widetilde{P}\widetilde{E} + \widetilde{E_0}\widetilde{Q})\} & (\bar{C} + \bar{S}G\bar{H})^T & (\widetilde{P}\widetilde{E} + \widetilde{E_0}\widetilde{Q})^T\widetilde{B} \\ * & -\gamma I & 0 \\ * & * & -\gamma I \end{bmatrix} < 0, \tag{35}$$

which is rewritten as

$$\widetilde{\Omega} + \widetilde{\Gamma}G\widetilde{\Pi} + (\widetilde{\Gamma}G\widetilde{\Pi})^T < 0, \tag{36}$$

where

$$
\widetilde{\Omega} = \begin{bmatrix} sym\{\bar{A}^T(\widetilde{P}\widetilde{E}+\widetilde{E_0}\widetilde{Q})\} & \bar{C}^T & (\widetilde{P}\widetilde{E}+\widetilde{E_0}\widetilde{Q})^T\widetilde{B} \\ * & -\gamma I & 0 \\ * & * & -\gamma I \end{bmatrix},
$$

$$
\widetilde{\Gamma} = \begin{bmatrix} (\widetilde{P}\widetilde{E}+\widetilde{E_0}\widetilde{Q})^T\bar{F} \\ \bar{S} \\ 0 \end{bmatrix}, \widetilde{\Pi} = \begin{bmatrix} \bar{H} & 0 & 0 \end{bmatrix}.
$$

Based on Lemma 4, the LMI in (36) has a solution $G$ if and only if the following two inequalities hold,

$$
\widetilde{\Gamma}^{\perp}\widetilde{\Omega}\widetilde{\Gamma}^{\perp T} < 0, \tag{37}
$$

$$
\widetilde{\Pi}^{T\perp}\widetilde{\Omega}\widetilde{\Pi}^{T\perp T} < 0, \tag{38}
$$

where

$$
\widetilde{\Gamma}^{\perp} = \begin{bmatrix} [I\ 0] & 0 & 0 \\ [0\ 0] & 0 & I \end{bmatrix} \begin{bmatrix} (\widetilde{P}\widetilde{E}+\widetilde{E_0}\widetilde{Q})^{-T} & 0 & 0 \\ 0 & I & 0 \\ 0 & 0 & I \end{bmatrix},
$$

$$
\widetilde{\Pi}^{T\perp} = \begin{bmatrix} [C_1\ 0] & 0 & 0 \\ [0\ 0] & I & 0 \\ [0\ 0] & 0 & I \end{bmatrix}, \quad C_1 = (C^T)^{\perp}.
$$

Then (37) and (38) are equivalent to

$$
\begin{bmatrix} AR + R^T A^T & B \\ * & -\gamma I \end{bmatrix} < 0, \tag{39}
$$

$$
\begin{bmatrix} C_1 sym\{A(PE+E_0Q)\}C_1^T & C_1 L^T & C_1(PE+E_0Q)^T B \\ * & -\gamma I & 0 \\ * & * & -\gamma I \end{bmatrix} < 0. \tag{40}
$$

Set $P_1 = P - P_{12}P_{22}^{-1}P_{21}$, $P_2 = P_{12}P_{22}^{-1}P_{21}$; then from (34), $R^{-1} = P_1 E + E_0 Q$, $P = P_1 + P_2$, (39) and (40) are equivalent to (23) and (24), $rank(P_2) \leq \hat{n}$. The proof is completed.

**Remark 1.** *Theorem 1 gives a necessary and sufficient condition for solving the reduced-order $H_\infty$ filtering problem of singular FOS. The inequalities in (23) and (24) are LMIs, while the constraint $rankP_2 \leq \hat{n}$ has to resort to a numerical algorithm based on alternating projections in [28], which is used later for solving the same problems; see for instance [12] and [30].*

**Remark 2.** *When $\alpha = 1$, Theorem 1 reduces to the reduced-order $H_\infty$ filtering design for singular integer-order systems. In this special case, the necessary and sufficient condition is modified as that there exist matrices $P_1 > 0$ and $P_2 \geq 0$, $rankP_2 \leq \hat{n}$ satisfying (23) and (24), which is equivalent to Theorem 3 in [12]. This verifies that Theorem 1 includes the result in [12] as a special case.*

When $E = I$ in (1), we can obtain the following result for the reduced-order $H_\infty$ filtering problem of FOS below:

$$
\begin{cases} D^\alpha x(t) = Ax(t) + B\omega(t) \\ y(t) = Cx(t) \\ z(t) = Lx(t) \end{cases} \tag{41}
$$

**Corollary 1.** *Consider the FOS in (41). The reduced-order $H_\infty$ filtering problem of FOS is solvable by an $\hat{n}$-th order filter in the form (4) if and only if there exist matrices $P_1 \in \mathbb{P}_\alpha^{n \times n}$ and $P_1 + P_2 \in \mathbb{P}_\alpha^{n \times n}$, $rankP_2 \leq \hat{n}$ satisfying*

$$\begin{bmatrix} A^T P_1 + P_1^T A & P_1^T B \\ * & -\gamma I \end{bmatrix} < 0, \tag{42}$$

$$\begin{bmatrix} C_1 sym\{A(P_1 + P_2)\}C_1^T & C_1 L^T & C_1(P_1 + P_2)^T B \\ * & -\gamma I & 0 \\ * & * & -\gamma I \end{bmatrix} < 0, \tag{43}$$

*where $C_1 = (C^T)^\perp$.*

**Proof.** The proof is similar to that of Theorem 1, so we omit it. □

**Remark 3.** *We can deduce an equivalent form of Corollary 1. Notice that (42) and (43) are equivalent to*

$$\begin{bmatrix} A^T \bar{P}_1 + \bar{P}_1 TA & \bar{P}_1^T B \\ * & -\gamma^2 I \end{bmatrix} < 0, \tag{44}$$

$$\begin{bmatrix} C_1[sym\{A(\bar{P}_1 + \bar{P}_2)\} + L^T L]C_1^T & C_1(\bar{P}_1 + \bar{P}_2)^T B \\ * & -\gamma^2 I \end{bmatrix} < 0, \tag{45}$$

*where $\bar{P}_1 \in \mathbb{P}_\alpha^{n \times n}$ and $\bar{P}_1 + \bar{P}_2 \in \mathbb{P}_\alpha^{n \times n}$, $rank\bar{P}_2 \leq \hat{n}$. Multiplying (42) by $\gamma$, left- and right-multiplying (43) by $diag(\gamma^{1/2} I, \gamma^{-1/2} I, \gamma^{1/2} I)$, then applying Schur complement, we deduce the equivalent form by setting $\bar{P}_1 = \gamma P_1$, $\bar{P}_2 = \gamma P_2$. When $\alpha = 1$, Corollary 1 reduces to the reduced-order $H_\infty$ filtering design for integer-order systems. In this special case, the necessary and sufficient condition is modified such that there exist matrices $\bar{P}_1 > 0$ and $\bar{P}_1 + \bar{P}_2 > 0$, $rank\bar{P}_2 \leq \hat{n}$ satisfying (44) and (45), which is equivalent to the condition of Theorem 1 in [29] with the case $D = 0$. Corollary 1 is an extension of Theorem 1 in [29].*

Now, we study the zeroth-order $H_\infty$ filtering problem of singular FOS and give the following result.

**Theorem 2.** *The zeroth-order $H_\infty$ filtering problem for the singular FOS in (1) is solvable with a filter in the form (8) if and only if there exist matrices $P \in \mathbb{P}_\alpha^{n \times n}$ and $Q$ satisfying*

$$\begin{bmatrix} sym\{A^T(PE + E_0 Q)\} & (PE + E_0 Q)^T B \\ * & -\gamma I \end{bmatrix} < 0, \tag{46}$$

$$\begin{bmatrix} C_1 sym\{A(PE + E_0 Q)\}C_1^T & C_1 L^T & C_1(PE + E_0 Q)^T B \\ * & -\gamma I & 0 \\ * & * & -\gamma I \end{bmatrix} < 0. \tag{47}$$

*where*

$$C_1 = (C^T)^\perp, \quad E^T E_0 = 0. \tag{48}$$

*With the solutions $P$ and $Q$, the parameter in (8) is given by*

$$\hat{D} = -S^{-1}\Gamma^T \Delta \Pi^T \Lambda + \Sigma^{1/2} W \Lambda^{1/2} \tag{49}$$

*where S, W, Λ, Δ, and Σ satisfy (20)–(22) with parameters Ω, Γ, Π as follows.*

$$\Omega = \begin{bmatrix} sym\{A^T(PE + E_0Q)\} & L^T & (PE + E_0Q)^T B \\ * & -\gamma I & 0 \\ * & * & -\gamma I \end{bmatrix},$$

$$\Gamma = \begin{bmatrix} 0 \\ -I \\ 0 \end{bmatrix}, \quad \Pi = \begin{bmatrix} C & 0 & 0 \end{bmatrix}. \tag{50}$$

**Proof.** Letting $\tilde{z}(t) = z(t) - \hat{z}(t)$, the filtering error system can be described as

$$\begin{cases} ED^\alpha x(t) = Ax(t) + B\omega(t) \\ \tilde{z}(t) = (L - \hat{D}C)x(t) \end{cases} \tag{51}$$

By Lemma 3, the filtering error system in (51) is admissible and the transfer function $G_{\omega\tilde{z}}(s)$ of the error system satisfies $\|G_{\omega\tilde{z}}(s)\|_\infty < \gamma$ if and only if there exist matrices $P \in \mathbb{P}_\alpha^{n \times n}, Q$ such that

$$\begin{bmatrix} sym\{A^T(PE + E_0Q)\} & (L - \hat{D}C)^T & (PE + E_0Q)^T B \\ * & -\gamma I & 0 \\ * & * & -\gamma I \end{bmatrix} < 0. \tag{52}$$

Furthermore, the above LMI can be separated as

$$\Omega + \Gamma\hat{D}\Pi + (\Gamma\hat{D}\Pi)^T < 0, \tag{53}$$

where

$$\Omega = \begin{bmatrix} sym\{A^T(PE + E_0Q)\} & L^T & (PE + E_0Q)^T B \\ * & -\gamma I & 0 \\ * & * & -\gamma I \end{bmatrix}, \Gamma = \begin{bmatrix} 0 \\ -I \\ 0 \end{bmatrix}, \Pi = \begin{bmatrix} C & 0 & 0 \end{bmatrix}. \tag{54}$$

Based on Lemma 4, (53) has a solution $\hat{D}$, if and only if the following two inequalities hold,

$$\Gamma^\perp \Omega \Gamma^{\perp T} < 0, \tag{55}$$
$$\Pi^{T\perp} \Omega \Pi^{T\perp T} < 0. \tag{56}$$

where

$$\Gamma^\perp = \begin{bmatrix} I & 0 & 0 \\ 0 & 0 & I \end{bmatrix}, \quad \Pi^{T\perp} = \begin{bmatrix} C_1 & 0 & 0 \\ 0 & I & 0 \\ 0 & 0 & I \end{bmatrix}, \tag{57}$$

then (55) and (56) are equivalent to (46) and (47). □

**Remark 4.** *Theorem 2 presents a necessary and sufficient condition for designing the zeroth-order $H_\infty$ filter of singular FOS with LMIs. When $\alpha = 1$, Theorem 2 reduces to the zeroth-order $H_\infty$ filtering design for integer-order systems. In this special case, the necessary and sufficient condition is modified such that there exists a matrix $P > 0$ satisfying (46) and (47), which is equivalent to Theorem 5 in [12].*

From Theorem 2, the following Corollary 2 is obviously true.

**Corollary 2.** *The zeroth-order $H_\infty$ filtering problem of FOS in (41) is solvable by a filter in the form (8) if and only if there exist matrix $P \in \mathbb{P}_\alpha^{n \times n}$ satisfying*

$$\begin{bmatrix} A^T P + P^T A & P^T B \\ * & -\gamma I \end{bmatrix} < 0, \tag{58}$$

$$\begin{bmatrix} C_1(AP + P^T A^T)C_1^T & C_1 L^T & C_1 P^T B \\ * & -\gamma I & 0 \\ * & * & -\gamma I \end{bmatrix} < 0, \tag{59}$$

*where $C_1 = (C^T)^\perp$.*

**Remark 5.** *Corollary 2 provides a necessary and sufficient condition for designing the zeroth-order $H_\infty$ filter of FOS. When $\alpha = 1$, Corollary 2 reduces to the zeroth-order $H_\infty$ filtering design for integer-order systems. In this special case, the necessary and sufficient condition is modified as that there exists a matrix $P > 0$ satisfying (58) and (59). Through a similar analysis of Remark 3, this result is equivalent to Theorem 3 in [29] with the case $D = 0$. Corollary 2 can be regarded as a generalization of Theorem 3 in [29].*

**Remark 6.** *In this paper, we study the problem of reduced-order $H_\infty$ filter design for commensurate fractional-order systems. We believe that the results are useful for further research on noncommensurate order systems.*

## 4. Illustrative Examples

In this section, we illustrate the validity of the proposed results by examples.

**Examples 1.** *Consider the system (1) with paramerters $\alpha = 0.7$, and*

$$E = \begin{bmatrix} 1 & 0 & 0 \\ 0 & 1 & 2 \\ 1 & 0 & 0 \end{bmatrix}, A = \begin{bmatrix} -1.5 & 0.8 & -2 \\ 5 & 0.7 & -4 \\ -3 & -1 & -2 \end{bmatrix}, B = \begin{bmatrix} 1 & 1 \\ -1 & -1 \\ 0 & 1 \end{bmatrix}, \tag{60}$$

$$C = \begin{bmatrix} 1 & 0 & -1 \end{bmatrix}, L = \begin{bmatrix} 0 & -1 & -1 \end{bmatrix}.$$

*In this example, we will design a filter such that the error system is admissible and the transfer function satisfises the given $H_\infty$ norm bound $\gamma = 0.56$. In order to use Theorem 1, we choose*

$$E_0 = \begin{bmatrix} 1 \\ 0 \\ -1 \end{bmatrix} \text{ and } C_1 = \begin{bmatrix} 1 & 0 & 1 \\ 0 & 1 & 0 \end{bmatrix}. \text{ By solving the inequalities in (23) and (24), we find}$$

*the solutions:*

$$P_1 = \begin{bmatrix} 7.5734 & 0.2275 & -5.9033 \\ -0.0982 & 0.0776 & 0.2717 \\ -6.8912 & -0.0360 & 7.4480 \end{bmatrix},$$

$$P_2 = \begin{bmatrix} 0.0188 & 0.0147 & -0.0365 \\ 0.0147 & 0.0116 & -0.0286 \\ -0.0365 & -0.0286 & 0.0707 \end{bmatrix},$$

$$Q = \begin{bmatrix} -0.7881 & -0.3386 & -0.2066 \end{bmatrix}.$$

*The reduced-order $H_\infty$ filtering problem for this singular FOS can be solved and rank $P_2 = 1$. For designing a 1-th $H_\infty$ filter, we can choose*

$$P_{12} = \begin{bmatrix} 0.97 & 0.76 & -1.88 \end{bmatrix}^T, \ P_{21} = P_{12}^T, \ P_{22} = 50,$$

and $P_{12}P_{22}^{-1}P_{21} = P_2$. *Then, we choose*

$$S = \begin{bmatrix} 0.0002 & 0 \\ 0 & 0.4079 \end{bmatrix},$$

*such that* $\Delta > 0$. *Furthermore, we have*

$$\Gamma = \begin{bmatrix} 0 & 0 & 0 & 0 & -1 & 0 & 0 \\ 0.97 & 0.76 & -1.88 & 50 & 0 & 0 & 0 \end{bmatrix}^T,$$

$$\Pi = \begin{bmatrix} 1 & 0 & -1 & 0 & 0 & 0 & 0 \\ 0 & 0 & 0 & 1 & 0 & 0 & 0 \end{bmatrix}.$$

*By Theorem 1, a desired reduced-order filter can be designed as*

$$\begin{cases} D^\alpha \hat{x}(t) = -1.1295\hat{x}(t) - 0.0850y(t) \\ \hat{z}(t) = -2.6280\hat{x}(t) - 0.0952y(t) \end{cases}$$

The above filter ensures that the error system is admissible and the transfer function meets the given $H_\infty$ norm bound $\gamma$.

**Examples 2.** *Consider the system* (1) *with paramerters* $\alpha = 0.8$, *and*

$$E = \begin{bmatrix} 1 & -2 & 0 \\ 0.5 & 1 & 0 \\ -2 & 3 & 0 \end{bmatrix}, A = \begin{bmatrix} 2 & 9 & -0.3 \\ 3 & -7 & 0 \\ 1.5 & -5 & -2 \end{bmatrix}, B = \begin{bmatrix} 1 & 0 \\ 1 & 1 \\ 0 & 1 \end{bmatrix}, \tag{61}$$

$$C = \begin{bmatrix} 1 & 0 & 1 \end{bmatrix}, L = \begin{bmatrix} 0 & -1 & 1 \end{bmatrix}.$$

*Given* $H_\infty$ *norm bound* $\alpha = 0.35$, *in order to use Theorem 2, we choose* $E_0 = \begin{bmatrix} 0 \\ 0 \\ -1 \end{bmatrix}$ *and*

$C_1 = \begin{bmatrix} 1 & 0 & -1 \\ 0 & 1 & 0 \end{bmatrix}$. *By solving the inequalities in* (46) *and* (47), *we find the solutions:*

$$P = \begin{bmatrix} 26.0588 & 6.7339 & 14.7876 \\ 6.6102 & 1.9608 & 3.9702 \\ 14.9447 & 4.3670 & 41.3467 \end{bmatrix},$$

$$Q = \begin{bmatrix} -65.8372 & 99.0206 & -0.4107 \end{bmatrix}.$$

*Choosing* $S = 0.0555$ *such that* $\Delta > 0$, *through calculation, we can get* $\hat{D} = -0.0803$,

$$\hat{z} = -0.0803y(t).$$

*Then by Theorem 2, the zeroth-order* $H_\infty$ *filtering problem for this singular FOS can be solved.*

## 5. Conclusions

This paper deals with the reduced-order $H_\infty$ filtering problem of singular FOS with fractional order $0 < \alpha < 1$. Based on the bounded real lemma for singular FOS, the necessary and sufficient conditions have been obtained without decomposing the system matrices. The method is in terms of two strict LMIs and a rank constraint, which can be solved by an alternating algorithm proposed in [29]. In particular, the method for solving the zeroth-order $H_\infty$ filtering problem is in terms of two strict LMIs. When reduced to special cases, our results are shown to be equivalent to those in the literature. Illustrative examples have been provided to explain the applicability of the proposed methods. Extensions to analysis and design problems for singular commensurate and noncommensurate FOS will be our future research work.

**Author Contributions:** Formal analysis, Y.G. and C.L.; Writing—review and editing, Y.G., C.L. and B.C. All authors have read and agreed to the published version of the manuscript.

**Funding:** This research was funded by National Natural Science Foundation of China: 61873137, 61973179 and Shandong Taishan Scholar Project: ts20190930.

**Conflicts of Interest:** The authors declare no conflict of interest.

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
