# Peer review of "Reduced-Order H∞ Filter Design for Singular Fractional-Order Systems"

_fractalfract, doi:10.3390/fractalfract6020097_

Round 1
Reviewer 1 Report
This paper investigates the problem of reduced-order H¥ filter design for singular fractional-order systems with order 0 < a < 1. This paper proposes a study of necessary and sufficient conditions for designs of both reduced-order H¥ filter and zeroth-order H¥ filter for singular fractional-order systems with order 0 < a < 1.
The paper is interesting and well presented. A minor revision could address the following points.
- The state of the art regarding the problem of fractional order system models and reduced order should be developed further, certain related papers should be addressed,
- Bourouba, S. Ladaci, A. Chaabi(2018) Reduced order model approximation of fractional order systems using Differential Evolution algorithm Journal of Control, Automation and Electrical Systems 29: 1. 32–43
- Saxena; V. Yogesh; P. P. Arya, Reduced-order modeling of commensurate fractional-order systems, 2016 14th International Conference on Control, Automation, Robotics and Vision (ICARCV), Phuket, Thailand, Nov, 2016.
- The authors should explain the purpose of this reduction of order: Why? And at what amount? What is the price to pay for such modification in terms of precision and quality?
- The main result seems to be well defined and demonstrated.
Reviewer 2 Report
Please see attached file.
